# Brain MRI and neuropsychological findings at long-term follow-up after COVID-19 hospitalisation: an observational cohort study

Lovisa Hellgren ![ORCID],[1] Ulrika Birberg Thornberg,[2] Kersti Samuelsson,[2] Richard Levi,[2] Anestis Divanoglou,[2] Ida Blystad[3,4]

For numbered affiliations see end of article.

**Correspondence to**
Lovisa Hellgren;
lovisa.hellgren@liu.se

## ABSTRACT

**Objectives** To report findings on brain MRI and neurocognitive function, as well as persisting fatigue at long-term follow-up after COVID-19 hospitalisation in patients identified as high risk for affection of the central nervous system.

**Design** Ambidirectional observational cohort study.

**Setting** All 734 patients from a regional population in Sweden with a laboratory-confirmed COVID-19 diagnosis admitted to hospital during the period 1 March to 31 May 2020.

**Participants** A subgroup (n=185) with persisting symptoms still interfering with daily life at a telephone follow-up 4 months after discharge were invited for a medical and neuropsychological evaluation. Thirty-five of those who were assessed with a neurocognitive test battery at the clinical visit, and presented a clinical picture concerning for COVID-19-related brain pathology, were further investigated by brain MRI.

**Main outcome measures** Findings on brain MRI, neurocognitive test results and reported fatigue.

**Results** Twenty-five patients (71%) had abnormalities on MRI; multiple white matter lesions were the most common finding. Sixteen patients (46%) demonstrated impaired neurocognitive function, of which 10 (29%) had severe impairment. Twenty-six patients (74%) reported clinically significant fatigue. Patients with abnormalities on MRI had a lower Visuospatial Index (p=0.031) compared with the group with normal MRI findings.

**Conclusions** In this group of patients selected to undergo MRI after a clinical evaluation, a majority of patients had abnormal MRI and/or neurocognitive test results. Abnormal findings were not restricted to patients with severe disease.

## Strengths and limitations of this study

► Detailed clinical characteristics and self-reported long-term outcomes for this cohort were previously presented as part of the Linköping COVID-19 Study.

► Participants were examined with MRI about 7 months after admittance to hospital and all MRI images were assessed by an experienced neuroradiologist.

► Neurocognitive tests and questionnaires were administered in a standardised way and evaluation of the data was performed by experienced neuropsychologists.

► The lack of a control group for this cohort was a study limitation.

► The unavailability of premorbid cognitive testing and MRI needs to be taken into consideration when interpreting the findings.

## INTRODUCTION

COVID-19 is an infectious disease primarily affecting the respiratory system, but there is accumulating evidence that this virus also affects the nervous system.[1,2] The exact mechanism behind neurological affection is yet to be elucidated, and whether symptoms depend on direct viral infection or secondary inflammatory effects remains debatable.[1]

Patients with COVID-19 undergoing imaging of the central nervous system (CNS) during the acute phase exhibit various pathological findings.[3] Brain MRI in the acute phase has shown abnormal findings, commonly infarctions, microscopic haemorrhages and/or intra-axial susceptibility abnormalities.[4,5] Lu *et al*[6] investigated 60 patients who had recovered from COVID-19 3 months after symptom onset. Changes in the microstructure of the white matter as well as in cortical areas pertaining to the olfactory system were reported.[6] To date, very little has been published regarding long-term MRI findings in the brain of patients with COVID-19.

In addition to visualisable pathology by brain MRI, COVID-19 has also been shown to lead to cognitive impairments in several domains, persisting several months after discharge.[7–9] It has been suggested that the severity of the infection (defined in terms of level of medical care needed[10,11] and/or levels of biomarkers of inflammation[7,9])

has a bearing on the degree of resulting neurocognitive impairment as well as findings from MRI.[12]

Our understanding of this disease is still incomplete, particularly regarding the risk for persisting neurocognitive symptoms and potential structural brain damage. The aims of this study were to report the association of brain MRI findings and neurocognitive function, as well as persisting fatigue at long-term follow-up after COVID-19 hospitalisation in patients identified as high risk for CNS affection.

## METHODS

This study forms part of the Linköping COVID-19 Study (LinCoS), an ambidirectional observational cohort study.[13] The reporting is informed by the Strengthening the Reporting of Observational Studies in Epidemiology statement for cohort studies.[14] LinCoS included all patients (n=734) with a laboratory-confirmed COVID-19 diagnosis admitted to hospital for COVID-19 in the total population of Region Östergötland, Sweden, during the period 1 March to 31 May 2020. LinCoS excluded cases with the following characteristics: (1) severe pre-existing comorbidities (such as dementia or under palliative care) making it impossible to establish any contribution of COVID-19 to neurocognitive dysfunction; (2) cause for hospitalisation unrelated to COVID-19 (such as an operation or labour), and where COVID-19 did not influence medical care, and thus could be deemed coincidental; and (3) those younger than 15 years of age.

All survivors at 4 months after discharge (n=460) were asked to participate in a structured telephone interview[13] led by one experienced rehabilitation professional (physician, neuropsychologist, occupational therapist or physiotherapist). The interview comprised 37 questions focusing on persisting COVID-19-related impairments and activity/participation limitations. The questions had the structure of yes/no, and for each yes, the patient was asked to estimate the impact on everyday life using a scale of 1–5. Individuals reporting symptoms such as muscle weakness, headache or cognitive problems significantly affecting their daily life (n=185) were then invited to a clinical evaluation, including a medical examination by a physician and a neuropsychological evaluation by a psychologist. Based on this clinical evaluation, individuals with concerning findings (on neurocognitive testing and/or events during acute hospitalisation indicating possible brain involvement) were referred to brain MRI. Thirty-five individuals who completed a valid neurocognitive assessment and underwent MRI were analysed as part of this study. A flow chart of the inclusion and exclusion process is shown in figure 1. The background data for patients at different stages of the inclusion process are presented in online supplementary table 1.

### Background data
#### Premorbid level of function
Premorbid level of function was determined based on patient's performance status and frailty 1 month preceding COVID-19, as reflected in the medical record in

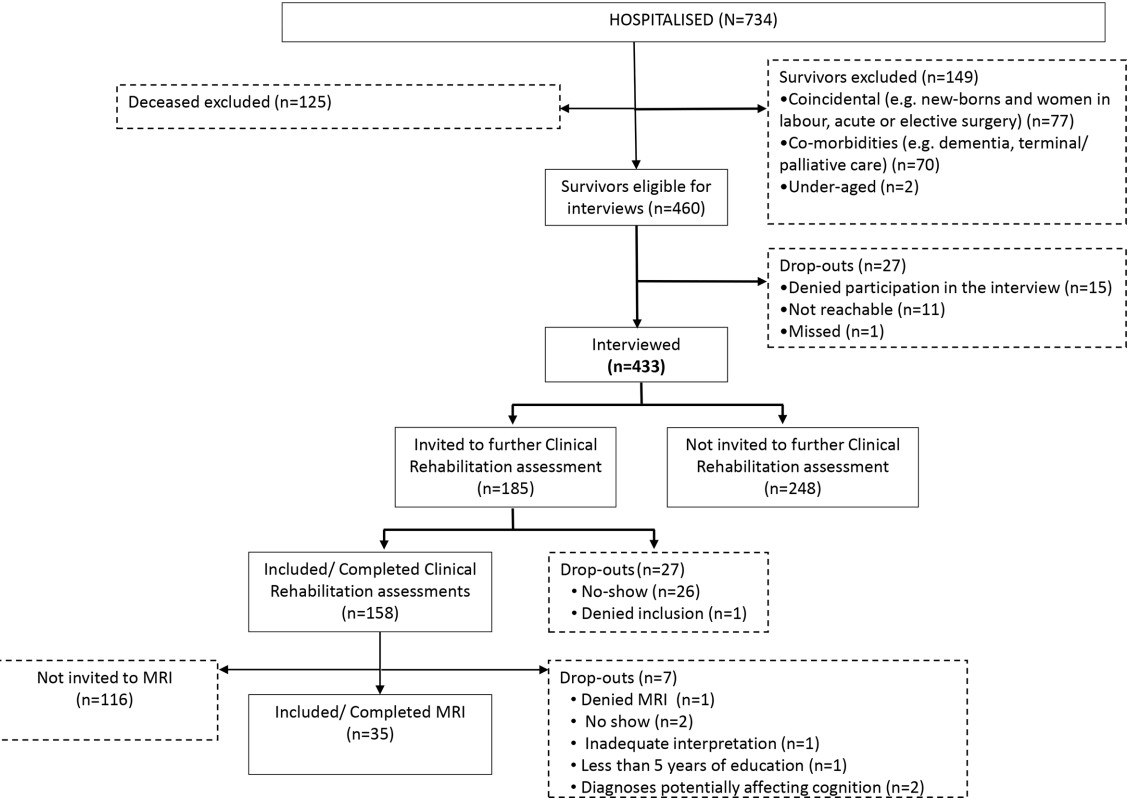

**Figure 1** Flow chart of the inclusion and exclusion process of patients previously hospitalised with COVID-19.

terms of employment, mobility and impact (type, severity and control) of comorbidities. This information was summarised in a modified version of the WHO/Eastern Cooperative Oncology Group (ECOG) Performance Status[15] and the Frailty Score according to Rockwood *et al*[16] to yield an assignment of the premorbid level of function in one of four groups:

1. No or mild frailty: no restriction in daily life activities.
2. Moderate frailty: person mobile and autonomous, but unable to perform physically demanding activities/work.
3. Considerable frailty: person usually able to perform basic daily life activities, but periodically confined to bed or chair.
4. Severe frailty: person permanently unable to perform activities of daily living and/or confined to bed or chair, for example, dementia necessitating care.

Individuals in group 4 were excluded from this study according to the LinCoS exclusion criteria described above.

## Severity of COVID-19

The WHO Clinical Progression Scale describes patient disease severity by tracking the patient trajectory and resource use over the course of clinical illness.[17] The highest WHO Clinical Progression Scale grade during hospitalisation and/or follow-up time (naming only the categories used in this study) was used:

4. Hospitalised: moderate disease, no oxygen therapy.
5. Hospitalised: moderate disease, oxygen by mask or nasal prongs.
6. Hospitalised: severe diseases, oxygen by non-invasive ventilation or high flow.
7. Hospitalised: severe diseases, intubation and mechanical ventilation, $pO_2/FiO_2 \geq 150$ or $SpO_2/FiO_2 \geq 200$.
8. Hospitalised: severe diseases, mechanical ventilation $pO_2/FiO_2 < 150$ ($SpO_2/FiO_2 < 200$) or vasopressors.
9. Hospitalised: severe diseases, mechanical ventilation $pO_2/FiO_2 < 150$ and vasopressors, dialysis or extracorporeal membrane oxygenation.

In addition, the maximum values of C-reactive protein (CRP) and D-dimer during hospitalisation were retrieved from medical records as additional proxy indicators of systemic disease severity.

## Educational level

Information about self-reported formal education was retrieved from the interview performed at 4 months after discharge:

1. Low level of formal education, ≤9 years of school.
2. Medium level of formal education, 9–12 years of school.
3. High level of formal education, >12 years of school.

## Self-estimated health

Data pertaining to self-estimated health before and after COVID-19 were obtained at the preceding screening interview at 4 months after discharge.[13]

## Outcome data

### Brain MRI

MRI was performed about 7 months (median 217 days; IQR 202–232 days) after admittance to hospital, and about 6.5 months (median 197 days; IQR 117–258 days) after discharge from hospital. MRI was performed on a clinical GE 3T Architect DV26 MR system according to the following clinical protocols: T2-Fluid Attenuated Inversion Recovery (T2-FLAIR) ax, T2-Fast Spin Echo (T2-FSE) ax, T1-FSE ax, Diffusion Weighted Imaging (DWI) ax, T2-FLAIR cor, 3D T1-Gradient Echo (T1-GRE) sag, susceptibility weighted image (SWI) ax. If performed, previous brain MRI or CT examinations were also assessed. MRI images were assessed by a neuroradiologist with 10 years of experience.

### Neuropsychological evaluation

Patients were assessed by psychologists with long experience from cognitive testing of patients with acquired brain injuries, using neurocognitive tests and questionnaires, approximately 5 months (median 142 days; IQR 120–167 days) after discharge from hospital. The time required for a neuropsychological evaluation was approximately 1 hour (range, 45–90 min). First, neurocognitive status was assessed using the Repeatable Battery for the Assessment of Neuropsychological Status (RBANS).[18] Then the patients completed the Hospital Anxiety and Depression Scale (HADS) and Multidimensional Fatigue Inventory (MFI).[19 20] In 4 of 35 patients, the neuropsychological evaluation was performed with the assistance of an authorised interpreter. The interpreters were trained in advance for the testing procedure to maintain compliance with the manual guidelines for each test. Only patients with reliable and valid test results were included, as identified through an individual case-by-case evaluation by members of the research team, resulting in one patient being excluded from interpreter non-compliant. Two patients were excluded due to premorbid diagnoses possibly affecting cognitive function. One patient was excluded due to having less than 5 years of formal education, thus not complying with the norm group for assessment of test results.[18]

RBANS is a validated battery for assessment of general neurocognitive status. The battery comprises 12 subtests, combined into five different indexes: Immediate Memory Index (List Memory and Story Memory subtests), Visuospatial Index (Figure Copy and Line Orientation subtests), Language Index (Picture Naming and Semantic Fluency subtests), Attention Index (Digit Span and Coding subtests) and Delayed Memory Index (List Recall, List Recognition, Story Recall and Figure Recall subtests).[18] RBANS total score is achieved by combining the scores of the five indexes. To make comparisons of test results across age spans, raw scores from the neuropsychological tests were transformed into index scores according to the test manual.[18] Each index score ranges from 40 to 160, with lower scores indicating lower performance. The population age-adjusted mean for each index is 100 with an SD of 15.[18] Test results are presented as the mean and SD, as well as frequencies of results below a cut-off (scores 2 SD below the mean according to norms).

To determine if the neurocognitive status should be considered impaired, the procedure suggested by Girard et al[21] and adapted for RBANS by Mitchell et al[22] was used. Cognition was considered as severely impaired when the patient scored 2 SD below the mean according to norms in at least two RBANS indexes, or 1.5 SD below the mean according to norms on at least three RBANS indexes. Cognition was considered as mildly/moderately impaired when the patient scored at least 2 SD below the mean according to norms on one of the RBANS indexes or 1.5 SD below the mean according to norms on two RBANS indexes. When the patient scored higher than stated above, cognition was considered not impaired.

HADS[19] is a valid[19 23] and commonly used questionnaire to detect depression or anxiety. HADS includes 14 questions, categorised into two subscales: depression and anxiety. Each question is scored from 0 to 3, giving a maximum score of 21 for each subscale. In this study, the cut-off for potential cases of depression/anxiety was ≥8, as suggested by Bjelland et al.[23]

The Swedish version of MFI is a valid questionnaire that includes 19 questions focusing on different aspects of fatigue: general fatigue, physical fatigue, mental fatigue, reduced motivation and reduced activity.[20] Each question is scored from 1 to 5 giving a total score ranging from 19 to 95, with higher scores indicating higher levels of fatigue. In this study, the cut-off for clinically significant fatigue was ≥53, as suggested by Hinz et al.[24]

## Statistical analysis

Descriptive results are presented as the mean and SD or median and IQR, as well as frequency. Fisher's exact test was used to compare differences between groups, except for RBANS indexes when Student's t-test was used. Spearman's correlation coefficient ($r_s$) was used for correlation analyses. In addition, analysis of variance was used to assess for differences in background data. All statistical tests were two sided and the accepted level of significance was $p<0.05$. The reported p values were not adjusted for multiple comparisons due to the increased likelihood of a type II error.[25] Data were analysed using IBM SPSS Statistics V.27.

## RESULTS

A description and comparison of the patients included in the study, subdivided into patients with abnormal versus normal MRI findings, are presented in table 1.

**Table 1** Characteristics of all patients included in the study, subdivided into those with abnormal MRI/normal MRI findings and previously hospitalised with COVID-19

| | All patients (n=35) | Patients with abnormal MRI (n=25) | Patients with normal MRI (n=10) | P value |
|---|---|---|---|---|
| Age, median (IQR) years | 59 (51–66) | 62.0 (56.0–69.0) | 50.5 (47.75–56.5) | 0.007 |
| Men/women, n*† | 28/7 | 21/4 | 7/3 | n.s. |
| Days in hospital, median (IQR)† | 18 (7–47) | 25 (7–55) | 18.0 (8.75–23) | n.s. |
| ICU care, need/no need, n* | 20/15 | 13/12 | 7/3 | n.s. |
| Mechanical ventilation, need/no need, n* | 19/14 | 13/12 | 6/4 | n.s. |
| Premorbid function category 1/2/3/4, n* | 20/12/3/0 | 11/11/3/0 | 9/1/0/0 | 0.046 |
| Self-estimated previous health, n* | 13/12/9/0/0 | 9/9/7/0/0 | 4/3/2/0/0 | n.s. |
| Self-estimated present health, n* | 4/4/18/8/0 | 3/4/15/3/0 | 1/0/3/5/0 | n.s. |
| WHO Clinical Progression Scale 4/5/6/7/8/9, n* | 4/8/4/0/6/13 | 2/7/3/0/4/9 | 2/1/1/0/2/4 | n.s. |
| Educational level 1/2/3, n* | 10/15/10 | 9/10/6 | 1/5/4 | n.s. |
| CRP median (IQR)† | 202 (144–322) | 223 (139–329) | 176.5 (119.5–289.5) | n.s. |
| D-dimer median (IQR)† | 0.72 (0.38–2.15) | 0.76 (0.56–2.28) | 0.56 (0.29–1.95) | n.s. |

Categories of premorbid function: (1) No or mild frailty, no restriction in daily life; (2) Moderate frailty, mobile and independent, but unable to handle physically demanding activities or work; (3) Considerable frailty, ability to perform activities of daily living, but in periods confined to bed or chair; (4) Severe frailty, not able to perform activities of daily living and/or confined to bed or chair. Dementia necessitating care. WHO Clinical Progression Scale: (4) Hospitalised, moderate disease, no oxygen therapy; (5) Hospitalised, moderate disease, oxygen by mask or nasal prongs; (6) Hospitalised, severe diseases, oxygen by non-invasive ventilation or high flow; (7) Hospitalised, severe diseases, intubation and mechanical ventilation, $pO_2/FiO_2 \geq 150$ or $SpO_2/FiO_2 \geq 200$; (8) Hospitalised, severe diseases, mechanical ventilation $pO_2/FiO_2 <150$ ($SpO_2/FiO_2 <200$) or vasopressors; (9) Hospitalised, severe diseases, mechanical ventilation $pO_2/FiO_2 <150$ and vasopressors, dialysis or ECMO. Categories of educational level: (1) Up to 9 years of school; (2) 9–12 years of school; (3) More than 12 years of school. Categories of self-estimated health (previous and present): (1) Excellent; (2) Very good; (3) Good; (4) Fair; (5) Poor.
*Fisher's test for comparison of frequencies.
†Median test.
CRP, C-reactive protein; ECMO, extracorporeal membrane oxygenation; ICU, Intensive Care Unit; n.s., not significant.

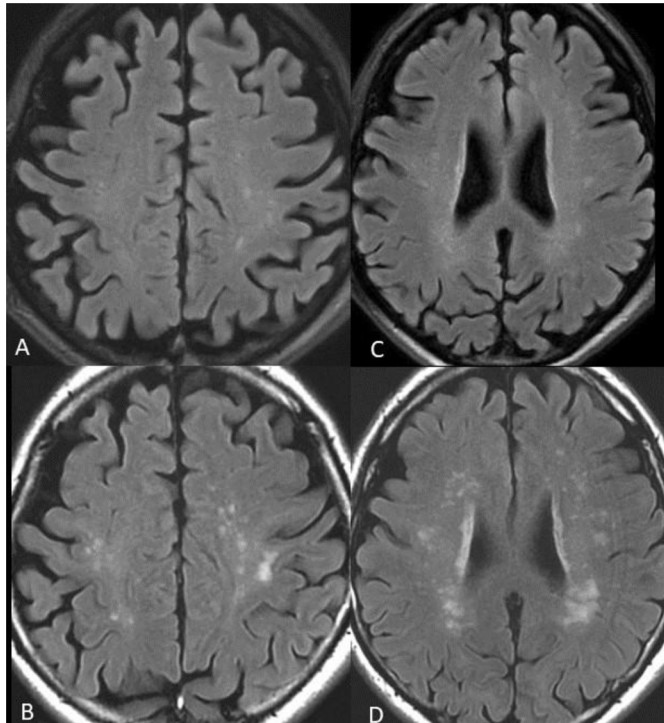

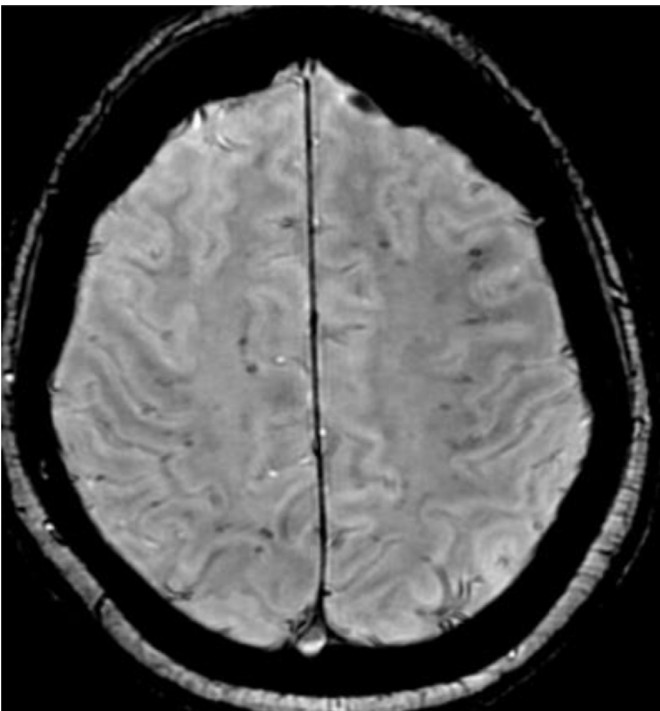

**Figure 2** T2-FLAIR images of one of the patients. (A) and (C), superior row, are from the MRI examination performed during the acute phase. (B) and (D), inferior row, are from the MRI examination at late follow-up. New white matter lesions appear at the follow-up MRI, with a typical distribution with a predilection for the parietal and frontal lobes. T2-FLAIR, T2-Fluid Attenuated Inversion Recovery.

**Figure 3** Susceptibility weighted image (SWI) of one of the patients at follow-up MRI. There are multiple SWI abnormalities, mainly located in the grey-white matter junction.

### MRI findings

Twenty-five patients (71%) showed multiple subcortical white matter lesions, located in the cerebral hemispheres near the grey-white matter junction, particularly in the frontal and parietal lobes. Six patients had had MRI performed in the acute phase during their hospitalisation, and all of these patients had additional white matter lesions at the follow-up MRI. Figure 2 shows an example of a patient with new white matter lesions at follow-up, with a typical distribution.

Seven of the patients with pathological MRI had confluent white matter lesions, with a propensity for a biparietal distribution, whereas eight patients had abnormalities on the SWI, mostly located subcortically, near the grey-white matter junction. Figure 3 shows an example of the SWI pattern in one of the patients. The MRI findings are summarised in online supplementary table 2.

### Neuropsychological findings

Sixteen of 35 patients (46%) showed cognitive impairments; 6 of these (17%) showed mildly/moderately impaired cognition, and 10 patients (29%) had severely impaired cognition according to Mitchell's definition.[22] Results below cut-off were apparent in all cognitive domains. Immediate Memory and Delayed Memory were the indices where most patients performed below cut-off (2 SD below the population-based mean) (table 2).

RBANS total score index did not correlate with either age or time in hospital. Further, there were no significant differences between RBANS total scores regarding WHO Clinical Progression Scale categories, need for care in Intensive Care Unit or mechanical ventilator, premorbid level of function categories, the self-estimated health before hospitalisation, the level of education or the need for an interpreter. Cases with missing data in some subtests of RBANS did not differ in background characteristics or outcomes compared with the cases with complete tests (online supplementary table 3).

Clinically significant fatigue (MFI ≥53) was reported by 25 patients (74%) (table 3). The median scores on the HADS subscales for depression and anxiety were 3.0 (IQR 1.5–5.0) and 4.0 (IQR 2.0–6.5), respectively. Depression (cut-off ≥8) was reported by five patients (14%) and anxiety (cut-off ≥8) by six patients (17%).

### MRI findings in relation to neuropsychological findings

Patients were divided into two groups depending on whether there were any MRI findings (abnormal MRI) or not (normal MRI). Demographic and background data for the two groups are presented in table 1. Patients with abnormal MRI were older (p=0.007) and had a lower premorbid level of function (p=0.046). There were no between-group differences in days in hospital, need for ICU/mechanical ventilation, CRP or D-dimer levels, or WHO Clinical Progression Scale categories (table 1). The mean value in the Visuospatial Index was lower in the group with abnormal MRI (mean 81.8; SD 15.1) compared with the group with normal MRI (mean 94.3; SD

**Table 2** RBANS results of patients previously hospitalised with COVID-19

|  | n[I] | Mean (SD) | Below cut-off, n[II] (%) |
|---|---|---|---|
| Immediate Memory Index | 32 | 89.2 (21.2) | 8 (25) |
| Visuospatial Index | 33 | 85.2 (15.1) | 4 (12) |
| Language Index | 35 | 90.9 (16.0) | 5 (14) |
| Attention Index | 34 | 87.1 (21.4) | 3 (9) |
| Delayed Memory Index | 32 | 83.9 (18.1) | 6 (19) |
| RBANS total score | 31 | 83.0 (17.8) | 8 (26) |

Frequencies of results below cut-off (scores at least 2 SD below the age-adjusted population mean of 100, SD 15 according to the norms) are presented.
n[I], the number of patients who completed subtests needed for the index; n[II], the number of patients with results below cut-off; RBANS, Repeatable Battery for the Assessment of Neuropsychological Status.

11.3) (p=0.031). Otherwise, there were no between-group differences regarding neurocognition, fatigue, depression or anxiety.

## DISCUSSION

This study presents brain MRI and neurocognitive findings, as well as self-reported fatigue at long-term follow-up after COVID-19 hospitalisation in patients with clinical risk indicators of higher cerebral dysfunction. The main radiological finding was the presence of multiple white matter lesions with a frontal and parietal distribution. This is in line with previous findings on MRI performed during the acute phase of COVID-19, where lesions associated with microvascular structures in the white matter have been reported.[5] White matter lesions are often found with increasing age and can be a marker of cerebrovascular disease. In the present study, the group with abnormal MRI were older, which might be a partial explanation for white matter lesions being more frequent in that group. However, a meta-analysis of neurological complications associated with COVID-19 also found that cerebrovascular disease was the most common neurological

**Table 3** Results of MFI on patients previously hospitalised with COVID-19 (n=34)

| Subscales of MFI | Median (IQR) |
|---|---|
| General Fatigue (max 20) | 15 (11–18) |
| Physical Fatigue (max 20) | 14 (1.5–18) |
| Mental Fatigue (max 15) | 9 (6–12) |
| Reduced Activity (max 20) | 13 (11.5–16.25) |
| Reduced Motivation (max 20) | 10.5 (8–13.25) |
| MFI full score (max 95) | 63 (49.75–70) |

MFI, Multidimensional Fatigue Inventory; n, number of patients who completed the questionnaire.

injury in individuals aged >50 years, occurring in more than 50% of individuals, and could not be explained by increasing age.[26] This suggests that there seems to be an age-associated vulnerability for cerebrovascular disease as a complication of COVID-19 infection. Furthermore, all the patients who underwent MRI during the acute phase of COVID-19 had acquired multiple new white matter lesions when examined at follow-up. This suggests that the COVID-19 infection might have an impact on the brain, even in the aftermath of the acute and subacute virus infection.

Fatigue was reported by 74% (n=25) of the patients. This is in line with previous reports on fatigue being a common long-term symptom following COVID-19.[27–29] The median score on the MFI subscale General Fatigue at 5 months' follow-up was 15 (IQR 11–18), which should be compared with mean scores at 4 months' follow-up on patients with traumatic brain injury ranging from 10.7 to 12.5 depending on the severity of the traumatic brain injury.[30]

Every fourth patient (29%, n=10) had neurocognitive test results corresponding to a severe impairment, whereas half of the patients (54%, n=19) did not present any neurocognitive impairment. Because there were no differences regarding cognition (RBANS total scale score) between the patients with different levels of self-estimated previous health, premorbid function or level of education, the results might indicate that the impaired cognition detected was caused by the current period of illness rather than any background factors. These findings imply that, in some individuals, a COVID-19 infection might have a negative impact on cognition lasting at least several months after discharge from hospital.

Immediate Memory and Delayed Memory were the indices where most patients scored below cut-off, a finding in agreement with previous research showing impaired results for memory related to COVID-19 infection.[11] Because memory, as well as fatigue, seems to be affected by COVID-19, future research should explore possible interactions with cognition and fatigue following COVID-19.

Few studies have reported associations between MRI findings and neuropsychological functioning following COVID-19.[12] Our results showed that the mean value for the Visuospatial Index was significantly lower in the group with abnormal MRI compared with the group with normal MRI. The relevance of this result is difficult to interpret. The p value was not corrected for multiple comparisons, the sample size was small and only four patients (12%) were below the cut-off demarcating impairment. On the other hand, most patients with abnormal MRI had subcortical white matter lesions, and white matter abnormalities have been shown to be associated with impairment in several cognitive domains.[31 32] The parietal lobes are considered essential for visuospatial functioning, because several neural pathways important for visuospatial processing pass the parietal lobes[33] and visuospatial functions have been attributed to bilateral parietal activation.[34] Thus, the finding in the abnormal MRI group of white matter lesions with a biparietal distribution

might be related to the low visuospatial test results seen in this group.

However, there was no significant difference between the two MRI groups regarding the frequency of patients with cognitive impairments or with clinically significant fatigue. This suggests that neither fatigue nor a cognitive impairment after a COVID-19 infection is necessarily captured by presence of abnormal MRI findings in the brain. Thus, when assessing patients with persistent neuropsychological complaints following COVID-19, we suggest a multimodal approach including neurocognitive assessment and close evaluation of potential need for MRI.

It has been suggested that there may be an association between days on ventilator care and MRI findings in the brain of patients with COVID-19 in the acute phase.[35] In this cohort, the presence or absence of MRI abnormalities was not associated with ICU care or mechanical ventilation. The inclusion of patients for MRI in this study might have affected this result, and since this group had a more severe illness compared with the whole cohort of LinCoS there is a possibility that the used inclusion criteria and the lack of control group reduced the reliability of the conclusion that MRI abnormalities were unrelated to disease severity. However, MRI abnormalities were found both in patients who had received ICU care, as well as in patients who did not need this level of care. SWI abnormalities have been reported in the acute phase of COVID-19 infection.[5] In this cohort, SWI abnormalities were found in 8 of 35 cases, which is less frequent than reported from the Stockholm cohort published in 2020,[5] where 29 of 39 patients had this finding. However, in the Stockholm cohort study, most of the patients had been on a ventilator in the ICU, whereas in the present cohort, only 54% of the patients needed ventilator care, thus indicating less severe illness in our cohort.

In the present study, there were no significant correlations between RBANS total scale score, time in hospital or scores on the WHO Clinical Progression Scale. This finding is in contrast to previous studies which suggested that the severity of COVID-19 infection has a bearing on the degree of cognitive impairment.[10 11] In the Almeria *et al*'s study, early cognitive testing performed at 10–35 days after discharge showed a worse cognitive performance among patients who required oxygen therapy during hospitalisation.[11] In the present study, where neurocognitive assessment was performed 5 months after discharge, no such association was found.

Results from the present study suggest that MRI findings and/or neurocognitive impairments may occur also in patients with less severe COVID-19, and future research should therefore also include evaluation of patients with COVID-19 who have either been treated in regular pandemic wards or who have not been hospitalised at all.

## Strengths and limitations

The patients were carefully selected from a population-based cohort study including all confirmed cases of COVID-19 admitted to hospital during a 3-month period. Brain MRIs and neurocognitive assessments were analysed by an experienced neuroradiologist and two experienced neuropsychologists, respectively. RBANS is considered a comprehensive test battery[36] and has been recognised as the gold standard neurocognitive battery for diagnosis and clinical trial outcome measurement in mild cognitive impairment.[37] However, RBANS is a relatively brief battery and as such does not cover all aspects of cognition. The absence of premorbid cognitive assessment makes it hard to refute that any premorbid factor might have affected the results. However, when considering the background data available on the premorbid level of functioning and educational level, the poor performance on neurocognitive testing is unexpected in this cohort. In conjunction with frequent patient reports of new impairments in higher cerebral function, this points to the likelihood that these results indeed represent COVID-19-related problems.

With the lack of premorbid MRI, as well as a relatively small sample size, and because it was not manageable to include a control group, it is difficult to confirm that white matter changes could be solely attributed to the COVID-19 infection. However, in cases where an MRI was performed during the acute phase of the hospitalisation, all patients subsequently developed new white matter lesions, making it plausible to assume that the COVID-19 infection contributed to these white matter lesions. This will have an impact on consideration of differential diagnoses when evaluating follow-up MRI on patients who have had COVID-19, especially in younger patients who display white matter lesions on brain MRI after a COVID-19 infection.

## CONCLUSIONS

A majority of patients with concerning neuropsychological symptoms and/or medical history selected to undergo MRI after a clinical evaluation showed signs of possible COVID-19-related brain affection, detectable by brain MRI and/or neurocognitive test results. Deviating findings were not restricted to patients with severe disease. Thus, post-COVID-19-related changes could be considered in the differential diagnosis when white matter lesions are detected on brain MRI. Furthermore, it is important to consider post-COVID-19-related changes when dealing with patients' reports of neuropsychological deficiency, regardless of the severity of disease. As neuropsychological impairments are not always associated with abnormal brain MRI findings, a multiprofessional approach would be preferable when assessing patients with persisting neuropsychological complaints following COVID-19.

**Author affiliations**

[1]Department of Rehabilitation Medicine in Jönköping, Jönköping Region, and Department of Health, Medicine and Caring Sciences, Linköping University, Linköping, Sweden

[2]Department of Rehabilitation Medicine in Linköping, and Department of Health, Medicine and Caring Sciences, Linköping University, Linköping, Sweden

[3]Department of Radiology in Linköping and Department of Health, Medicine and Caring Sciences, Linköping University, Linköping, Sweden

[4]Centre for Medical Image Science and Visualization (CMIV), Linköping University, Linköping, Sweden

**Acknowledgements** We thank Lars Valter at Forum Östergötland, Faculty of Medicine and Health Sciences, Linköping University, Linköping, Sweden, for statistical advice; and Ninnie Håkansson at the Radiology Department of Linköping University Hospital for invaluable assistance with MRI bookings. Last, a special acknowledgement to Agnes Andersson, Julia Eriksson, Jacob Lennartsson, Malin Lindh and Mollie Rönn Holmström for valuable assistance with performing the neurocognitive evaluations.

**Contributors** LH, KS, AD, RL, UBT and IB took part in planning the study. LH, UBT and IB participated in data collection. LH, IB, UBT and KS analysed the data. LH and IB wrote the original draft of the manuscript. LH, KS, AD, RL, UBT and IB reviewed and edited the manuscript. LH is the guarantor of the study.

**Funding** The study was funded by the ALF grant.

**Disclaimer** The funder of the study had no role in study design, data collection, data analysis, data interpretation or writing of the report.

**Competing interests** None declared.

**Patient and public involvement statement** Formally, due to the early stage of the COVID-19 pandemic in Sweden, neither patients nor the public were involved in the design or planning of this study. We intend to seek patient and public involvement in the development of an appropriate method of dissemination. Results from the study will be presented to the public through public service. The patient organisation for COVID-19 in Sweden will be contacted and the results from the study will be shared. In addition, the results will be presented at relevant medical conferences.

**Patient consent for publication** Not required.

**Ethics approval** The study has been approved by the Swedish Ethical Review Authority (Dnr 2020-03029 and 2020-04443).

**Provenance and peer review** Not commissioned; externally peer reviewed.

**Data availability statement** No data are available. The Public Access to Information and Secrecy Act in Sweden prohibits us from making individual level data publicly available, thus no additional data will be made available.

**ORCID iD**
Lovisa Hellgren http://orcid.org/0000-0003-4099-9456

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
