## [Reviewer comments · BMJ Open]

ARTICLE DETAILS

TITLE (PROVISIONAL)	Brain MRI and neuropsychological findings at long-term follow-up after COVID-19 hospitalisation: an observational cohort study
AUTHORS	Hellgren, Lovisa; Birberg Thornberg, Ulrika; Samuelsson, Kersti; Levi, Richard; Divanoglou, Anestis; Blystad, Ida

VERSION 1 – REVIEW

REVIEWER	Martelletti, Paolo Sapienza University of Rome
REVIEW RETURNED	28-Jul-2021

GENERAL COMMENTS	The Authors present an interesting neuroradiological and neuropsychological study on 734 patients affected by COVID during the first wave and hospitalized. The structure of the study is valid, results are in line with the long-COVID widely described in literature. It would be useful to extract from the database the long-COVID headache, a very relevant clinical sign for any sequelae. I would be more careful in affirming that multiple white matter changes are directly correlated to COVID, since literature is widely oriented towards the many origins of these lesions and their presence also in healthy population. I would insist more on the neuropsychological impairment. In order to accomplish this concern, I suggest to insert and discuss the following papers: PMID: 33050880 PMID: 32965838 PMID: 32972360 PMID: 34036244
--

REVIEWER	Waiter, Gordon University of Aberdeen, Aberdeen Biomedical Imaging Centre
REVIEW RETURNED	05-Aug-2021

GENERAL COMMENTS	This is a timely and well written paper describing the neurological and psychological effects at long-term follow up of COVID-19 survivors. The authors present clear evidence of the presence of a range of imaging observations and cognitive test scores outside that expected norms. Specific comments: The authors describe the group with "abnormal" MRI findings as being significantly older. Without pre-COVID scanning it is not possible to attribute this abnormality to COVID rather than being normal age related increase in neuropathology. The presence of white matter hyperintensities on FLAIR images is part of normal
---

	ageing and is not therefore abnormal just by its presence. This partially addressed in the Discussion but should be expanded. In Supplementary Table 2 the authors use the term ".. changes on MRI" for those without previous MRI. This should be changed to something more appropriate.
--	---

REVIEWER	Menon, David University of Cambridge, Division of Anaesthesia, Department of Medicine
REVIEW RETURNED	05-Aug-2021

GENERAL COMMENTS	Hellgren et al report on cognitive deficits and MRI findings in a subset of 35 patients followed up after hospitalisation for acute COVID-19. They found that 25 of the patients had changes on conventional structural and susceptibility-weighted MRI (they did not report on diffusion weighted imaging), which had progressed in a subset of six patients who were also imaged acutely. 16 patients showed neurocognitive impairments on the RBANS scale, 10 of them severe. Cognitive deficits were unrelated to acute disease severity or any premorbid factors. Patients with MRI abnormalities were older and were assessed to have "lower premorbid function" (though the basis for this is unclear - was it the Rockwood score?), but had not suffered more severe Covid-19. The Visuospatial Index (one of six indices assessed by the RBANS) was lower in patients with MRI abnormalities. The results are interesting, but I have some questions about whether the study design is appropriate for addressing the relationship of MR changes to symptoms at follow up - in particular, the absence of Covid-19 controls (i.e. those without any symptoms at follow up). I also have some queries about study power and statistical reporting. I would ask the authors to respond to the following issues: Although the initial number of patients at follow up is reasonable (n= 734), the detailed basis for selection of a subgroup of 185 patients for follow up is unclear. In order to understand the population being assessed, it would be important to have a clear understanding of what symptoms qualified the 185 patients for recall versus the 248 who were not invited. Similarly, it is unclear to me what resulted in patients being invited for MRI (n= 35) versus the 116 who were not The basis for inclusion at each stage of the study could be usefully supplemented by data that show comparison of demographics and disease severity between the 35 patients eventually included in the analysis, versus the larger groups at each stage of filtering. The differences in Visuospatial Index between MRI positive and negative patients are reported with a p value of 0.031. Was this corrected for multiple comparisons?
---

	The lack of MRI data in patients who were asymptomatic at screening makes it difficult to understand whether the MRI findings are incidental, or truly related to the presence of cognitive dysfunction The small sample size makes it difficult to interpret the relevance of negative results. Similarly, if there was no correction for multiple comparisons, this should engender caution in interpreting positive results for correlations between imaging and cognitive, neuropsychological and functional metrics.
--	---

VERSION 1 – AUTHOR RESPONSE

Reviewers comments	Responses to comments	Revised pages
Reviewer: 1 It would be useful to extract from the database the long-COVID headache, a very relevant clinical sign for any sequelae.	While headache is indeed a relevant symptom in post-COVID syndrome, it was not the focus of the current manuscript. Given the limited space, we chose not to include this topic. However, data on headache as well as other reported symptoms of the 433 interviewed patients are presented in detail in the LinCoS main study, which is referred to in this paper.	
I would be more careful in affirming that multiple white matter changes are directly correlated to COVID, since literature is widely oriented towards the many origins of these lesions and their presence also in healthy population. In order to accomplish this concern, I suggest to insert and discuss the following papers: PMID: 33050880 PMID: 32965838 PMID: 32972360 PMID: 34036244	This study does not primarily report headache (see our previous answer), the 3 suggested references on COVID-19 and headache have not been inserted. Regarding the suggested paper on white matter lesions, it seems too general to be inserted in the context of this paper. It is common knowledge for radiologists that white matter lesions on MRI is a descriptive term, where the underlying cause can have many different origins. We have	Page 12 and 14

	expanded the discussion on the white matter lesions detected in the manuscript, and added a new reference in this section: PMID:34408638.	
I would insist more on the neuropsychological impairment.	Clarifying information have been inserted in the results section regarding neuropsychological data.	Page 10
Reviewer: 2 The authors describe the group with "abnormal" MRI findings as being significantly older. Without pre-COVID scanning it is not possible to attribute this abnormality to COVID rather than being normal age related increase in neuropathology. The presence of white mater hyperintensities on FLAIR images is part of normal ageing and is not therefore abnormal just by its presence. This partially addressed in the Discussion but should be expanded.	This is an important question to address, and the discussion has been expanded on the matter and a new reference has been added (PMID: 34408638).	Page 12
In Supplementary Table 2 the authors use the term ".. changes on MRI" for those without previous MRI. This should be changed to something more appropriate.	The terms have been changed from "changes" to "lesions" in the Table and in the manuscript.	Supplemental Table 2
Reviewer: 3 Patients with MRI abnormalities were older and were assessed to have "lower premorbid function" (though the basis for this is unclear - was it the Rockwood score?), but had not suffered more severe Covid-19.	As stated in the background data section: the premorbid level of function was determined based on the patient's performance status and frailty 1 month preceding COVID-19, summarized in a modified version of the WHO/ECOG Performance Status and the Frailty Score according to Rockwood.	
Although the initial number of patients at follow up is reasonable (n= 734), the detailed basis for selection of a subgroup of 185 patients for follow up is unclear. In order to understand the population being assessed, it would be important to have a clear understanding of what symptoms qualified the 185 patients for recall versus the 248 who were not invited.	The selection of patients is mainly described in the flowchart, but clarifications have been made in the Methods section regarding the role of the interview in the selection, as well as the decision on who to invite for an MRI, a decision based on the clinical evaluation by the	Page 4 and Supplemental Table 1.

Similarly, it is unclear to me what resulted in patients being invited for MRI (n= 35) versus the 116 who were not. The basis for inclusion at each stage of the study could be usefully supplemented by data that show comparison of demographics and disease severity between the 35 patients eventually included in the analysis, versus the larger groups at each stage of filtering.	physician and the psychologist. In addition, as suggested, Supplementary Table 1 has been included to describe background data for the groups.	
The differences in Visuospatial Index between MRI positive and negative patients are reported with a p value of 0.031. Was this corrected for multiple comparisons?	We have expanded the discussion on the relevance of this particular result. The p value of the difference in the Visuospatial Index between patients with normal/abnormal MRI has not been corrected for multiple comparisons. The risk of making a type 1 error is limited because the MRI shows a distribution of white matter lesions in locations known to be important for visuospatial function.	Page 13
The lack of MRI data in patients who were asymptomatic at screening makes it difficult to understand whether the MRI findings are incidental, or truly related to the presence of cognitive dysfunction.	We agree; this is mentioned as a study limitation, unfortunately a control group wasn't manageable.	
The small sample size makes it difficult to interpret the relevance of negative results.	We agree with the reviewer and have therefore added a comment about sample size in the Discussion as a study limitation.	Page 14

VERSION 2 – REVIEW

REVIEWER	Waiter, Gordon University of Aberdeen, Aberdeen Biomedical Imaging Centre
REVIEW RETURNED	14-Sep-2021
GENERAL COMMENTS	The authors have addressed my concerns adequately.

REVIEWER	Menon, David University of Cambridge, Division of Anaesthesia, Department of Medicine
REVIEW RETURNED	25-Sep-2021

GENERAL COMMENTS	The authors have provided additional data in response to my review, and I complement them on quickly collating the data that enabled production of Supplemental Table 1. I have some remaining concerns. The first of these is relatively straightforward - I had asked whether they had corrected the analysis of association between MRI and clinical, cognitive and other variables for multiple comparisons. They have not specified whether they did this, and without that information it is impossible to assess the significance of the reported p value of 0.031 for a lower Visuospatial Index in the group with MRI abnormalities. They need to explicitly state whether this is or is not corrected for multiple comparisons, and if it was not, state what the corrected p value was (or state that it was not significant, if that was the case) The second issue is a bit more complex, and arises from the data they now present in Supplemental Table 1. As the table shows, the patients who underwent MRI were clearly more severely ill, as defined by the number of days in hospital, requirement for critical care, distribution of WHO Progression Scale scores, and possibly premorbid function. The inference is that the MRI group selected out patients with more severe disease. This, in itself, is not critical - but it does diminish the chance of finding correlations between disease severity and MRI findings. The absence of controls who did not have the symptoms that resulted in imaging further compounds this confound. I think this issue needs to be explicitly acknowledged, since it reduces the reliability of the conclusion that MRI abnormalities were unrelated to disease severity.
---

VERSION 2 – AUTHOR RESPONSE

Reviewers comments	Responses to comments
I have some remaining concerns. The first of these is relatively straightforward - I had asked whether they had corrected the analysis of association between MRI and clinical, cognitive and other variables for multiple comparisons. They have not specified whether they did this, and without that information it is impossible to assess the significance of the reported p value of 0.031 for a lower Visuospatial Index in the group with MRI abnormalities. They need to explicitly state whether this is or is not corrected for multiple comparisons, and if it was not, state what the corrected p value was (or state that it was not significant, if that was the case)	The reported p-values were not adjusted for multiple comparisons, due to the increased likelihood of a type II error. We have made clarifications about this both in the results section (page 8) and the discussion section (page 12-13).
The second issue is a bit more complex, and arises from the data they now present in Supplemental Table 1. As the table shows, the patients who underwent MRI were clearly more severely ill, as defined by the number of days	We agree that the selection of patients might affected this results

in hospital, requirement for critical care, distribution of WHO Progression Scale scores, and possibly premorbid function. The inference is that the MRI group selected out patients with more severe disease. This, in itself, is not critical - but it does diminish the chance of finding correlations between disease severity and MRI findings. The absence of controls who did not have the symptoms that resulted in imaging further compounds this confound. I think this issue needs to be explicitly acknowledged, since it reduces the reliability of the conclusion that MRI abnormalities were unrelated to disease severity.

and have clarified this in the discussion section (page 13).